# Concurrent *TP53* and *CDKN2A* Gene Aberrations in Newly Diagnosed Mantle Cell Lymphoma Correlate with Chemoresistance and Call for Innovative Upfront Therapy

**DOI:** 10.3390/cancers12082120

**Published:** 2020-07-31

**Authors:** Diana Malarikova, Adela Berkova, Ales Obr, Petra Blahovcova, Michael Svaton, Kristina Forsterova, Eva Kriegova, Eva Prihodova, Lenka Pavlistova, Anna Petrackova, Zuzana Zemanova, Marek Trneny, Pavel Klener

**Affiliations:** 1First Department of Internal Medicine-Hematology, General University Hospital and First Faculty of Medicine, Charles University in Prague, 12808 Prague, Czech Republic; diana.tuskova@vfn.cz (D.M.); adela.berkova@vfn.cz (A.B.); blahovcova@lymphoma.cz (P.B.); kristina.forsterova@lf1.cuni.cz (K.F.); trneny@cesnet.cz (M.T.); 2Institute of Pathological Physiology, First Faculty of Medicine, Charles University in Prague, 12853 Prague, Czech Republic; 3Center of Oncocytogenomics, Institute of Clinical Biochemistry and Laboratory Diagnostics, General University Hospital and First Faculty of Medicine, Charles University in Prague, 12853 Prague, Czech Republic; eva.prihodova@vfn.cz (E.P.); Lenka.Pavlistova@vfn.cz (L.P.); zuzana.zemanova@vfn.cz (Z.Z.); 4Department of Hemato-Oncology, Faculty of Medicine and Dentistry, Palacky University and University Hospital Olomouc, 77515 Olomouc, Czech Repulbic; Ales.Obr@fnol.cz (A.O.); Eva.Kriegova@fnol.cz (E.K.); anna.petrackova@gmail.com (A.P.); 5CLIP, Department of Pediatric Hematology/Oncology, Second Faculty of Medicine and University Hospital in Motol, 15006 Prague, Czech Republic; michael.svaton@lfmotol.cuni.cz

**Keywords:** mantle cell lymphoma, TP53, CDKN2A, prognostic markers, chemoresistance

## Abstract

Mantle cell lymphoma (MCL) is a subtype of B-cell lymphoma with a large number of recurrent cytogenetic/molecular aberrations. Approximately 5–10% of patients do not respond to frontline immunochemotherapy. Despite many useful prognostic indexes, a reliable marker of chemoresistance is not available. We evaluated the prognostic impact of seven recurrent gene aberrations including tumor suppressor protein P53 (*TP53*) and cyclin dependent kinase inhibitor 2A (*CDKN2A*) in the cohort of 126 newly diagnosed consecutive MCL patients with bone marrow involvement ≥5% using fluorescent in-situ hybridization (FISH) and next-generation sequencing (NGS). In contrast to *TP53*, no pathologic mutations of *CDKN2A* were detected by NGS. *CDKN2A* deletions were found exclusively in the context of other gene aberrations suggesting it represents a later event (after translocation t(11;14) and aberrations of *TP53,* or ataxia telangiectasia mutated (*ATM*)). Concurrent deletion of *CDKN2A* and aberration of *TP53* (deletion and/or mutation) represented the most significant predictor of short EFS (median 3 months) and OS (median 10 months). Concurrent aberration of *TP53* and *CDKN2A* is a new, simple, and relevant index of chemoresistance in MCL. Patients with concurrent aberration of *TP53* and *CDKN2A* should be offered innovative anti-lymphoma therapy and upfront consolidation with allogeneic stem cell transplantation.

## 1. Introduction

Mantle cell lymphoma (MCL) represents a subtype of non-Hodgkin lymphoma with poor prognosis, especially for patients who are resistant to front-line treatment [1,2,3,4,5]. Besides the translocation t(11;14), several recurrent cytogenetic aberrations have been reported [6,7,8,9,10,11,12,13]. Approximately 5–10% of patients with newly diagnosed MCL fail to achieve response to upfront immunochemotherapy regimen. Despite many prognostic markers including MCL international prognostic index (MIPI), proliferation marker Ki-67, deletion or mutation of *TP53*, deletion of cyclin dependent kinase 2A (*CDKN2A*), or blastoid morphology, a reliable marker of chemoresistance is not available at diagnosis [14,15,16,17,18].

Based on the so far published data we selected seven candidate genes (*TP53*, *CDKN2A*, ataxia teleangiectasia mutated (*ATM*), B-cell lymphoma 2 (*BCL2*), oncogene *MYC*, retinoblastoma protein 1 (*RB1*) and cyclin-dependent kinase 4 (*CDK4*)), and analyzed their aberrations using fluorescent in-situ hybridization FISH in 126 consecutive patients with newly diagnosed MCL with bone marrow (BM) involvement ≥5%. In 113 patients with available diagnostic DNA samples, *TP53* and *CDKN2A* mutation status was analyzed by next-generation sequencing (NGS) approach. The analyzed cytogenetic aberrations were correlated with overall response rate and survival (event-free survival—EFS, overall survival—OS).

## 2. Patients and Methods

### 2.1. Patients’ Characteristics

An unselected cohort of 223 patients with newly diagnosed MCL was subject to analysis. Diagnostic samples obtained from 126 newly diagnosed MCL patients with BM infiltration ≥5% were analyzed by fluorescence in situ hybridization (FISH) between 1 January 2009 and 30 June 2018 at the First Department of Internal Medicine-Hematology, General University Hospital and First Faculty of Medicine, Charles University, Prague. The study was approved by University General Hospital Ethics Committee 63/16 from 22 June 2016. From these 126 patients, 113 were subject to mutational analysis of *TP53* and *CDKN2A* genes by next generation sequencing (NGS). In addition, we analyzed baseline characteristics and survival of other 97 patients with newly diagnosed MCL (from the same time period), who were not subject to FISH or NGS analysis due to low BM infiltration (<5%) or lack of material.

### 2.2. Fluorescence in Situ Hybridization (FISH)

Cytogenetic and FISH analyses were implemented in the Center of Oncocytogenomics, General University Hospital, Charles University in Prague, Czech Republic, accredited according to ISO 15189. Detailed protocols and a list of FISH probes used are available in the Appendix A.

### 2.3. TP53 and CDKN2A Mutation Assessment by Next-Generation Sequencing (NGS)

NGS was implemented at the University Hospital Olomouc with the regular control of laboratory performance using the internal standards of known mutation load (2% and 5% variant allele frequency-VAF, respectively) in each run and periodic inspection by an external agency. The full coding sequence of the *TP53* (exons 2–11 including 2 bp intronic overlap, plus 5′and 3′UTR; NM_000546) and *CDKN2A* (exons 1–3 including 2 bp intronic overlap, plus 5′and 3′UTR; NM_000077) were analyzed by targeted ultra-deep NGS as reported previously [19,20]. Amplicon-based libraries were sequenced as a paired-end on MiSeq (2 × 151 bp, Illumina, San Diego, CA, USA) with minimum target read depth of 5000x. The detection limit was set up to 1%, and mutations within the range 1–3% were confirmed by replication [20]. All detected variants were manually inspected using the Integrative Genomics Viewer (IGV) and annotated using variant population databases (1000 Genomes, gnomAD, ExAC), clinical mutation databases (ClinVar, COSMIC, IARC TP53 Database) and/or functional prediction tools (SIFT, PolyPhen-2). Only pathogenic or likely pathogenic variants were reported. Polymorphisms were filtered out using variant population databases (1000 Genomes, gnomAD, ExAC). The data can be downloaded as Appendix A from the journal site.

### 2.4. Statistical Analysis

Categorical data were compared by chi-square tests, numerical data with Mann–Whitney U tests. Kaplan–Meier survival analysis was used to estimate EFS and OS, and the statistical significance between survival curves was assessed by a log-rank test. Cox regression multivariate analysis was used to calculate the effect of the variables upon EFS and OS. Data were analyzed using GraphPad Prism version 7.00 for Windows, GraphPad Software, La Jolla, CA, USA, and open-source RStudio, Boston, MA, USA.

We used the Random Forests for Survival, Regression, and Classification R-package to perform all random forests analysis. Variable importance (VIMP) was estimated based on the effect of random permutations on the prediction error. To eliminate the variance of VIMP, all calculations were repeated 100 times and mean values for each VIMP obtained [21].

## 3. Results

### 3.1. Baseline Characteristics of the Analyzed Patients

We analyzed prognostic significance of molecular-cytogenetic aberrations of seven genes (*TP53*, *CDKN2A*, *ATM*, *BCL2*, *MYC*, *RB1,* and *CDK4*) using FISH in 126 consecutive patients with newly diagnosed MCL with BM involvement ≥5% between 1 January 2009 and 30 June 2018 at the Center of Oncocytogenomics, General University Hospital in Prague (Figure 1).

In addition to FISH, mutational analysis of *TP53* and *CDKN2A* genes by NGS was implemented in 113 (87%) patients with available DNA. Baseline characteristics of the analyzed patients are displayed in Table 1.

Because FISH was analyzed only on BM (with infiltration ≥5%) or peripheral blood, but not on formalin-fixed paraffin-embedded tissue sections, the analyzed (FISH) patients (*n* = 126) represented approx. 57% of all MCL patients (*n* = 223) diagnosed at the First Dept. of Internal Medicine-Hematology, General University Hospital and First Faculty of Medicine, Charles University, Prague (Figure 1). From the 97 patients (43%) with no available FISH data, 72 patients (74%) had undetectable or low (<5%) infiltration of the BM. Besides that, FISH data were unavailable for 25 patients (26%) (no diagnostic samples available) (Table 1). The applied methodology thus inevitably led to over-representation of high-risk patients according to MIPI in the cohort analyzed by FISH and NGS because patients with <5% infiltration of the BM were not analyzed. Indeed, while 5-year EFS and OS in the analyzed (FISH, NGS) cohort was 33.4 and 44.8%, respectively, 5-year EFS and OS of the non-analyzed cohort was 67.8% and 76.7%. (Figure 2). Median follow-up of the living patients reached 42 and 44 months in the analyzed and non-analyzed cohort, respectively.

### 3.2. Correlation of Baseline Clinical and Histopathological Parameters on Survival in the Cohort of 126 Patients with ≥5% BM Involvement

From the analyzed clinical and histopathological factors, the following parameters correlated with EFS and OS in the cohort of 126 patients with ≥5% BM involvement: MIPI (high risk vs. intermediate risk vs. low risk *p* < 0.0001), Ki-67 (≥30% vs. <30%), B symptoms (present vs. absent), complex karyotype (yes vs. no), type of therapy (intensified vs. R-CHOP-based vs. palliative—for EFS *p* = 0.0007 and for OS *p* = 0.0003). Splenomegaly and bulky disease (≥5 cm) correlated with EFS, but not OS (Appendix A). Despite the fact that all patients had infiltrated BM, the extent of BM infiltration positively correlated with shorter survival (for EFS, HR = 1.009 for each 1% of increase of BM infiltration, 95% CI = 1.001–1.02, *p* = 0.0312; for OS, HR = 1.016 for each 1% of increase of BM infiltration, 95% CI = 1.006–1.027, *p* = 0.002).

Deletion of CDKN2A can be detected almost exclusively in the context of other recurrently found cytogenetic aberrations

Distribution of the analyzed gene aberrations is displayed in Figure 3 and Appendix A.

Apart from *ATM*, all analyzed aberrations correlated with significantly shorter EFS as qualitative variables (i.e., aberration detected versus aberration not detected; Appendix A). Except for *ATM* and *MYC*, aberrations of the analyzed genes correlated with significantly shorter OS as qualitative variables (Appendix A). As few as 25 patients (19.8%) had no detectable gene aberration (apart from the translocation t(11;14)). A single (isolated) gene aberration was observed in 39 patients (31%), and ≥2 aberrations were detected in 62 patients (49.2%). Two, three, four, five, and six aberrations were detected in 22 (17.5%), 16 (12.7%), 16 (12.7%), 5 (4%), and 3 (2.4%) patients, respectively. From 39 patients with an isolated gene aberration, 15 patients had *TP53* gene aberration (11.9%), 9 patients had *ATM* deletion (7.1%), and 5 patients had *RB1* deletion (4%). Isolated gene aberrations of *BCL2* (*n* = 3, 2.4%), *MYC* (*n* = 3, 2.4%), *CDK4* (*n* = 3, 2.4%), and *CDKN2A* (*n* = 1, 0.8%) were rare. In addition, the only patient with isolated *CDKN2A* deletion had chromosome 9 monosomy (Figure 3, Appendix A). A Pearson chi-square test of seven analyzed aberrations revealed that *CDKN2A* gene deletion correlated with aberrations of *TP53*, *BCL2*, *RB1,* and *CDK4*, while aberration of *BCL2* correlated with *CDK4*. No other correlations were found among the seven analyzed genes (Table 2).

Distribution of the analyzed aberrations of 101 patients with at least one detected aberration including *TP53* mutation (except for the translocation t(11;14)). Each row represents one patient, gray squares represent aberrated genes, numbers represent type of aberration: 2 = monoallelic deletion (in case of *TP53* and/or mutation), 3 = bialellic deletion, 4 = monosomy, 5 = nullisomy, 6 = amplification, 7 = gain, 8 = trisomy, 9 = tetrasomy, 10 = *MYC* rearrangement; more numbers represent different subclones (e.g., 2,3 represent patients, in which both monoallelic and biallelic deletions of CDKN2A were detected).

*TP53* mutation and *TP53* deletion are both associated with adverse prognosis in MCL, while pathogenic *CDKN2A* mutations were not detected.

*TP53* gene aberration, either mutation, or deletion, was detected in 52 out of 126 patients (41.3%). Thirty-one of the analyzed patients (59.6%) with *TP53* aberration had both deletion and mutation of the *TP53* gene, while mutation without deletion, and deletion without mutation was detected in 12 (23.1%) and 5 (9.6%) patients. Another four patients (7.7%) had *TP53* deletion, but, due to lack of material, mutational analysis was not performed. Because *TP53* deletion and mutation significantly correlated with each other with respect to survival parameters (EFS and OS), we used *TP53* aberration (defined as *TP53* mutation and/or deletion) for all analyses (Appendix A). In addition to *TP53*, mutational analysis of *CDKN2A* gene was implemented by NGS, but no pathological mutations were identified.

A total number of gene aberrations is a strong predictor of outcome.

We confirmed that a complex karyotype significantly correlated with shorter survival (Figure 4A,B). Likewise, a total number of FISH aberrations (including *TP53* mutation) also correlated with EFS and OS. Interestingly, the biggest difference was observed between any two of the seven gene aberrations compared to any single (isolated) aberration, while three or more aberrations were not significantly worse predictors than two aberrations (Figure 4C,D). The total number of aberrations positively correlated with the male sex (Mann–Whitney U test *p* = 0.045).

Concurrent *TP53* aberration and *CDKN2A* deletion predicts chemoresistance.

We asked which of the two analyzed gene aberrations belonged to the most relevant pretreatment markers associated with the worst clinical outcome. First, Cox regression revealed that both *TP53* and *CDKN2A* aberrations independently correlated with shorter EFS and OS, while *BCL2* aberration correlated with shorter OS (Table 3). Second, a random forest analysis of the seven gene aberrations analyzed revealed that *CDKN2A* deletion is the most important predictor of short EFS and OS (Appendix A). Third, random forest analysis of all 21 possible aberration pairs unveiled that concurrent *TP53* gene aberration (*TP53*_mut/del_) and *CDKN2A* deletion (*CDKN2A*_del_) was the strongest predictor of short EFS, and together with concurrent *CDKN2A* and *BCL2* aberrations also predictor of short OS. (Appendix A).

Survival parameters of patients with concurrent *TP53*_mut/del_ and *CDKN2A*_del_ are shown in Figure 5.

Subanalysis of patients with concurrent *TP53*_mut/del_ and *CDKN2A*_del_ is given in Appendix A. The biggest differences between the patients with concurrent *TP53*_mut/del_ and *CDKN2A*_del_ and the remaining analyzed patients included frequency of CNS involvement (33% vs. 9%), B-symptoms (71% vs. 34%), MIPI (MIPI 3 in 79% vs. 58%) and Ki-67 ≥30% (85% vs. 40% in remaining patients). Only 38% and 17% of patients with concurrent *TP53*_mut/del_ and *CDKN2A*_del_ achieved response and complete response, respectively (compared to 79% and 56% of the patients without concurrent *TP53*_mut/del_ and *CDKN2A*_del_). At the time of database lock, 96% and 21% of patients with concurrent *TP53*_mut/del_ and *CDKN2A*_del_ had an event and were alive, respectively (compared to 54% and 65% in the remaining analyzed patients).

## 4. Discussion

As few as 25 patients (19.8%) had no detectable gene aberration (apart from the translocation t(11;14)). This is almost identical finding to that of Dalfau-Larue et al., who reported in their landmark analysis of 135 younger MCL patients from the European MCL Younger trial that only 24 patients (21%) displayed no copy number alteration (CNA) in any analyzed loci [18]. Delfau-Larue et al. analyzed CNA of similar genes to our own selection including *TP53*, *CDKN2A*, *ATM*, *RB1*, *CDK4*, and *MYC* (but not *BCL2*). Remarkably, the distribution of CNAs of the analyzed genes in the report of Dalfau-Larue was similar to our own data including *RB1* deletions (26% compared to 33%), *ATM* deletions (25% compared to 28%), *CDKN2A* deletions (25% compared to 33%), *TP53* deletions (22% compared to 32% of *TP53* gene deletions detected by FISH), *MYC* aberrations (18% compared to 26%), and *CDK4* gains (8% compared to 12%) (Figure 3, Appendix A). The overall lower incidence of all analyzed CNAs in the study of Delafau-Larue et al. compared to our own data can be explained by differences in the analyzed cohorts of patients. The significantly higher number of high-risk MIPI patients in our cohort (68%) compared to that of Delfau-Larue (25%) can be explained not only by different median age, but also by the fact that clinical trials usually do not enroll non-fit patients (slow go, no go) with poor performance status or serious comorbidities. In a recently published study by Wang et al., *MYC* rearrangements, but not *MYC* CNAs correlated with shorter OS independent of MIPI and Ki-67 [11]. In our study, 4 and 28 patients had *MYC* rearrangements and CNAs, respectively (one patient harbored both types of aberrations). While *MYC* aberrations correlated with shorter EFS in univariate analysis, its prognostic significance was lost in multivariate analysis (Table 3). In addition, *MYC* rearrangements were found exclusively in the context of other analyzed aberrations, suggesting it represents a later event similar to *CDKN2A* loss.

In the manuscript by Clot et al., 39 nodal MCL patients (62% belonging to high-risk MIPI group- exactly the same number as in our own analyzed cohort) were analyzed for *TP53* aberration and *CDKN2A* deletion (besides other analyzed genes) [17]. Curiously, 36% and 33% of the analyzed patients had *TP53* aberration and *CDKN2A* deletion, respectively (compared to 41% and 33% in our study). In addition, 13% patients had both *TP53* aberration and *CDKN2A* deletion (compared to 19% in our own study). Thus far, published results thus clearly confirm that both *TP53* aberrations and *CDKN2A* deletions are frequent in MCL, and that both are associated with adverse outcome.

### 4.1. TP53 Deletions and Mutations

Eskelund et al. recently reported that *TP53* mutations (detected in 20 patients, 11%) had significantly worse outcomes compared to *TP53* deletions (detected in 29 patients, 16%) [16]. In his report, only nine patients (47% patients from the *TP53* mutated cases, and 31% patients from the *TP53* deleted cases) had both *TP53* aberrations. In the study by Obr et al., 50% of patients with *TP53* deletions (with available sample for NGS) also had *TP53* mutations [19]. In our study, from 126 patients analyzed by FISH (for *TP53* deletion), 113 patients were also analyzed by NGS for *TP53* mutation. *TP53* aberration was observed in 52 patients (41.3% from 126), 40 patients had *TP53* deletion (31.7% from 126) and 43 patients had *TP53* mutation (38.1% from 113 analyzed by NGS). Thirty-one patients had both *TP53* aberrations (77.5% and 72.1% of the patients analyzed by FISH and NGS, respectively). Mutation without deletion of *TP53* was detected in 12 out of 43 patients (27.9%) analyzed by NGS. Deletion without mutation of *TP53* was detected in 9 out of 40 patients (22.5%) analyzed by FISH, but only in 5 out of 43 (11.6%) patients analyzed also by NGS (Appendix A). Interestingly, similar results have been reported for chronic lymphocytic leukemia (CLL) patients, where >70% of patients with *TP53* deletion also carried a *TP53* mutation [23]. Explanation for the observed differences between our data and the study of Eskelund et al. [16] might include different study cohorts (transplant-eligible patients included in clinical trials versus unselected, predominantly elderly patients with BM infiltration), and different methods for detection of *TP53* deletions (droplet digital polymerase chain reaction versus conventional FISH). Notably, in the study of Eskelund et al., patients with *TP53* mutation without deletion had a higher incidence of concurrent *CDKN2A* deletion (58%) compared to patients with *TP53* deletion without mutation (41%), which might at least partially explain the different prognostic impact of *TP53* mutation compared to deletion reported by the Nordic group.

### 4.2. Survival Analysis

Besides *TP53*, *CDKN2A* belongs to established prognostic markers in MCL [18]. *CDKN2A* deletion was observed in 41 out of 126 patients (32.5%) and represented the second most frequent aberration in the analyzed cohort. The data suggest that *CDKN2A* deletions represent late events in MCL because virtually all *CDKN2A* gene deletions were detected in the context of other genetic aberrations (Figure 3). Pearson’s analysis confirmed that *CDKN2A* correlated with incidence of all analyzed gene aberrations except for *ATM* deletion. In addition, *CDKN2A* deletions also correlated with male sex, MIPI, proliferation index by Ki-67, B-symptoms, CNS disease, and complex karyotype. The higher incidence of analyzed aberrations (especially *CDKN2A* and *RB1* deletions) observed in men might at least partially explain their worse outcome compared to women.

### 4.3. Concurrent Aberration of TP53 and Deletion of CDKN2A Is Associated with Chemoresistance

Concurrent *CDKN2A* deletion and *TP53* aberration were associated with chemoresistance to currently used upfront immunochemotherapy. We thus confirmed the findings of Delfau-Larue et al. on a real-life cohort of predominantly elderly MCL patients (median age 68 compared to 56 years) with significantly higher proportion of high-risk MIPI (62% compared to 25%). Subanalysis of these patients is given in Appendix A. Recently, Streich et al. reported that MCL with blastoid and pleomorphic morphology frequently harbor both *TP53* and *CDKN2A/B* aberrations, and that these cases are characterized by frequent chromothripsis [24]. Curiously, in sharp contrast to MCL, aberrations of *TP53* and *CDKN2A* were mutually exclusive in Burkitt lymphoma and were very rarely observed in diffuse large B cell lymphoma [25,26,27]. In indolent lymphoproliferative malignancies, namely CLL and follicular lymphoma, deletions of *CDKN2A* are rare, and *CDKN2A* inactivation frequently correlates with the transformation to an aggressive lymphoma with adverse prognosis [28,29,30,31]. From this perspective, MCL patients with concurrent *TP53*_mut/del_ and *CDKN2A*_del_ might be regarded as patients with “transformed” MCL and consequently with similarly adverse prognosis. Only three younger patients with concurrent *TP53*_mut/del_ and *CDKN2A*_del_ were successfully treated with salvage therapy and allogeneic stem cell transplantation (allo-SCT).

## 5. Conclusions

The molecular/cytogenetic index *TP53*_del/mut_ and *CDKN2A_del_* represents a novel, simple, and reliable pretreatment prognostic factor that identifies patients who do not profit from currently used therapies based on immunochemotherapy. Importantly, the combination of both aberrations represents a significantly more relevant prognostic marker of poor outcome compared to the isolated aberration of either gene. Patients with concurrent *TP53*_mut/del_ and *CDKN2A*_del_ might profit from innovative treatments, e.g., Bruton’s tyrosine kinase inhibitors, ideally in combination with other anti-lymphoma drugs (e.g., BH3-mimetics), and from upfront consolidation with allogeneic stem cell transplantation.

## Figures and Tables

**Figure 1 cancers-12-02120-f001:**
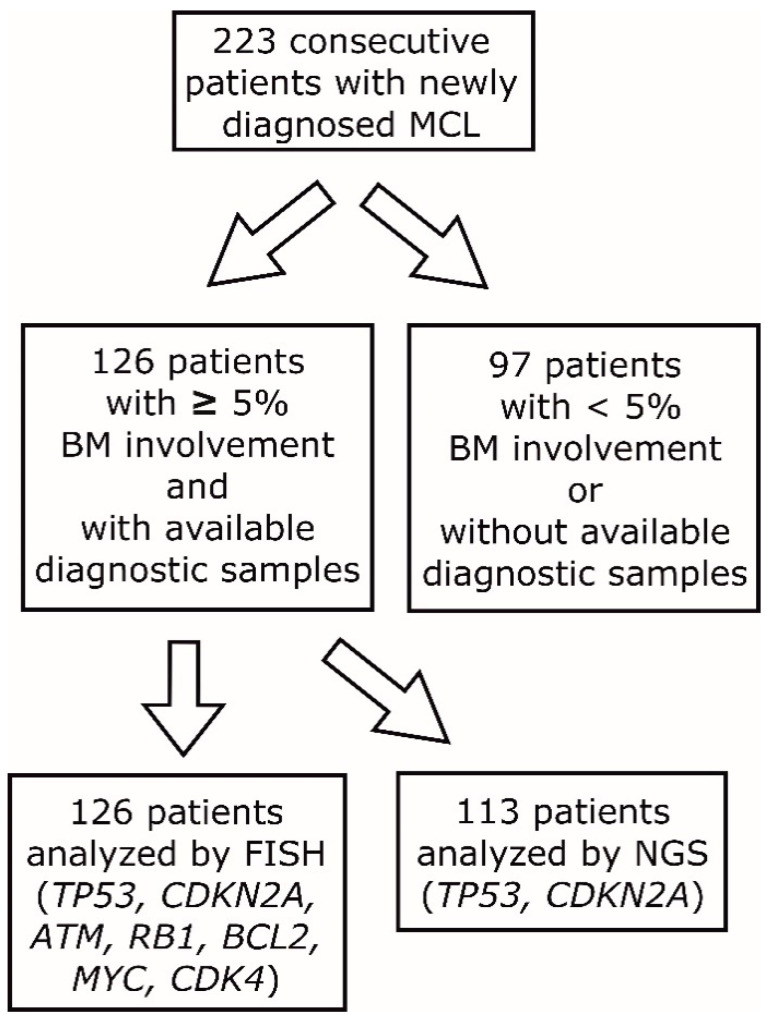
Flow chart of patient analysis. BM = bone marrow, FISH = fluorescent in-situ hybridization, NGS = next-generation sequencing.

**Figure 2 cancers-12-02120-f002:**
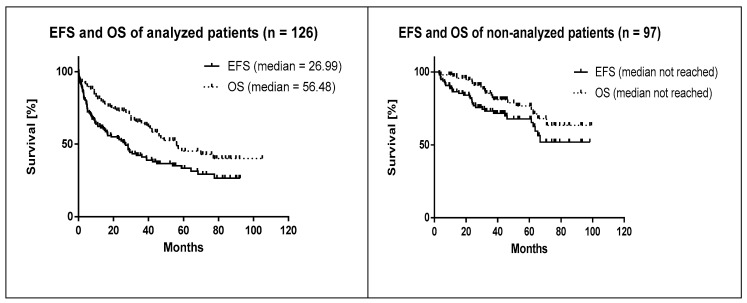
Survival in patients with bone marrow involvement ≥5% compared to patients without available diagnostic samples. EFS = event-free survival, OS = overall survival.

**Figure 3 cancers-12-02120-f003:**
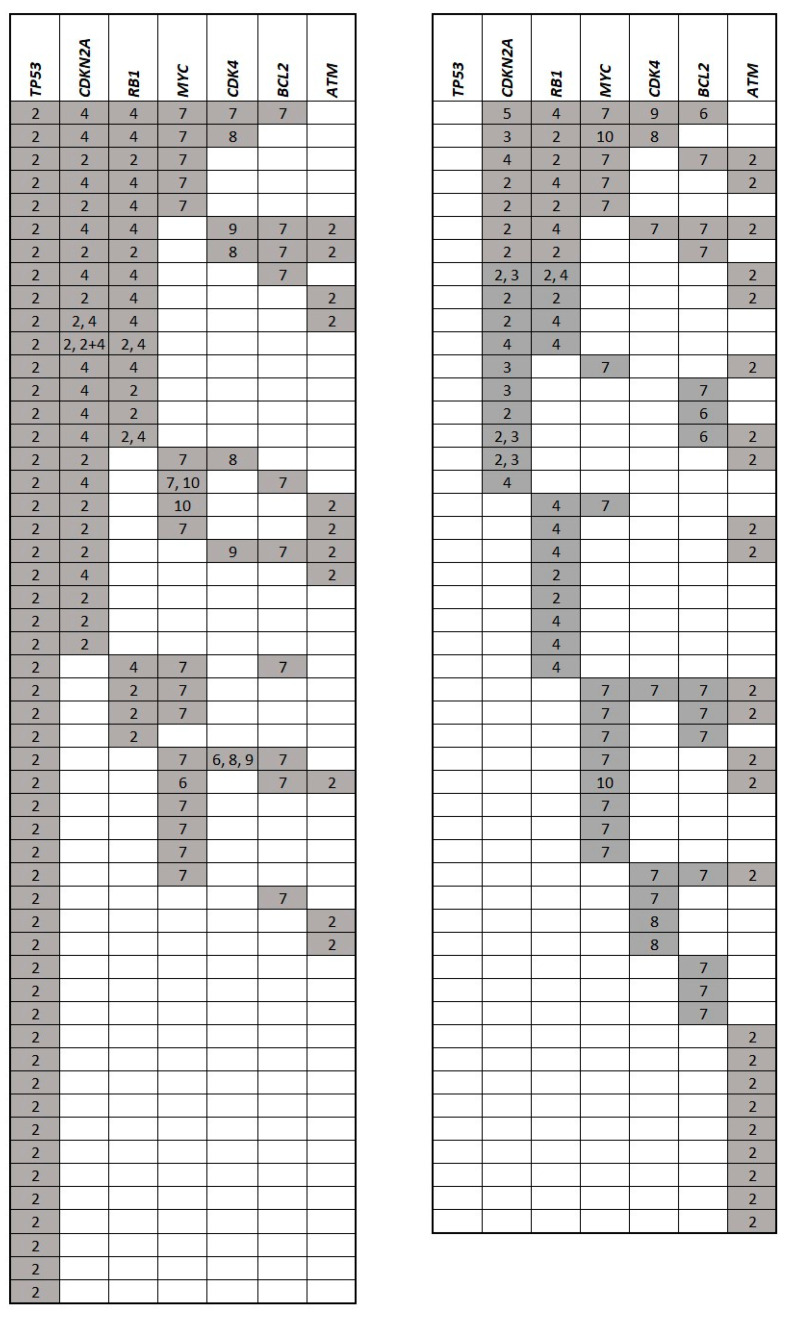
Distribution of the analyzed aberrations of 101 patients with at least one detected aberration including *TP53* mutation except for the translocation t(11;14). Each row represents one patient, gray squares represent aberrated genes, numbers represent type of aberration: 2 = monoallelic deletion (in case of *TP53* and/or mutation), 3 = bialellic deletion, 4 = monosomy, 5 = nullisomy, 6 = amplification, 7 = gain, 8 = trisomy, 9 = tetrasomy, 10 = *MYC* rearrangement.

**Figure 4 cancers-12-02120-f004:**
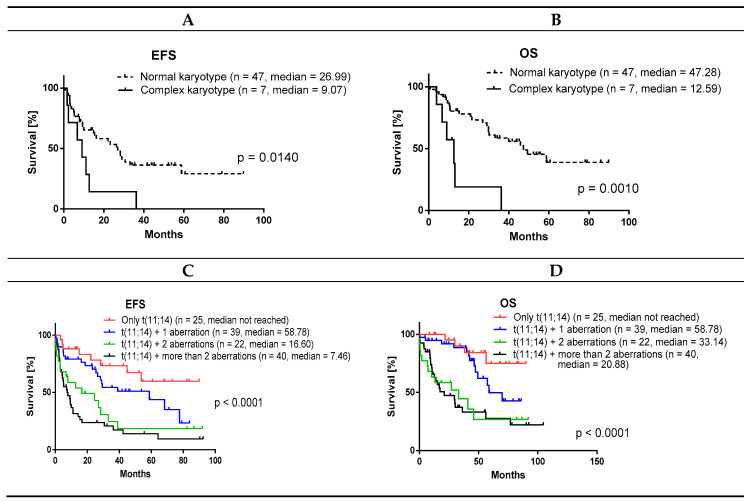
Complex karyotype and total number of gene aberrations correlates with shorter survival. (**A**,**C**) EFS; (**B**,**D**) OS. EFS = event-free survival; OS = overall survival.

**Figure 5 cancers-12-02120-f005:**
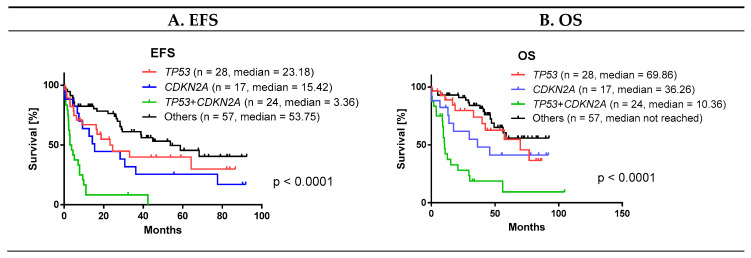
Patients with concurrent *TP53* and *CDKN2A* aberrations have significantly shorter survival than isolated aberrations. Patients in groups *TP53* and *CDKN2A* are patients with *TP53* aberrations or *CDKN2A* deletions that are not contained in the *TP53*+*CDKN2A* subcohort. (**A**) EFS, (**B**) OS. EFS = event-free survival, OS = overall survival.

**Table 1 cancers-12-02120-t001:** Baseline characteristics and response to therapy of the analyzed and non-analyzed patients.

Cohort	126 Patients with Bone Marrow Involvement ≥5%		97 Patients with No Available Diagnostic Samples	
Numbers (N) or percentages (%)	N	%	N	%
All patients	126	57	97	43
M	88	70	73	75
F	38	30	24	25
Age (median; years)	68		66	
Age (range; years)	29–82		40–87	
<65 years	47	37	44	45
≥65 years	79	63	53	55
Stage I–II	0	0	7	7
Stage III	0	0	13	13
Stage IV	126	100	77	79
Ki-67 ≥ 30% *	36	47	38	48
MIPI 1	19	15	25	26
MIPI 2	29	23	34	35
MIPI 3	78	62	38	39
B-symptoms	52	41	32	33
BM infiltration	126	100	73	75
BM infiltration ≥5%	126	100	25	26
Nodal involvement	108	86	88	91
Splenomegaly	89	71	46	47
Extra-hematological involvement	50	40	47	48
Bulky disease (≥5 cm)	45	36	27	28
CNS involvement **	17	13	7	7
Intensified therapy	37	29	50	52
R-CHOP-like therapy	71	56	38	39
Palliative therapy	8	6	7	7
Watch and wait	7	6	1	1
Died before initiation of therapy	3	2	0	0
Died during induction	9	7	0	0
ORR (CR/PR)	90	71	92	95
CR	61	48	73	75
PR	29	23	19	20
SD	4	3	3	3
PD	15	12	0	0
Event	78	62	32	33
Relapse	48	38	18	19
Death **	55	44	21	22

M = male; F = female; MIPI = MCL international prognostic index; BM = bone marrow; CNS = central nervous system; ORR = overall response rate; CR = complete remission; PR = partial remission; SD = stable disease; PD = progressive disease; R-CHOP = R(ituximab + C(yclopohosphamide) + H(ydroxydaunomycin) + O(ncovin) + P(rednisone); response was assessed by international workshop criteria published by Cheson et al. in 1999 [22]. * of the analyzed samples, ** anytime from diagnosis until database lock; differences >20% between cohorts are highlighted in gray.

**Table 2 cancers-12-02120-t002:** Correlation between the analyzed gene aberrations.

	*CDK4*	*RB1*	*BCL2*	*ATM*	*TP53*	*CDKN2A*	*MYC*
***CDK4***	1	0.138	**<0.001**	0.260	0.651	**0.016**	0.055
***RB1***		1	0.384	0.847	0.191	**<0.001**	0.074
***BCL2***			1	0.091	0.965	**0.012**	0.055
***ATM***				1	0.164	0.05	0.706
***TP53***					1	**0.006**	0.071
***CDKN2A***						1	0.065
***MYC***							1

The table shows *p*-values of Pearson’s chi-squared test. Statistically significant results are underlined and printed in bold fonts.

**Table 3 cancers-12-02120-t003:** Effect of the analyzed gene aberrations with survival parameters: multivariate analysis. (**A**) Event-Free Survival; (**B**) Overall Survival.

A. Event-Free Survival				B. Overall Survival			
Gene	HR	95% CI	*p*	Gene	HR	95% CI	*p*
*CDK4*	1.6	0.8–3.1	0.218	*CDK4*	1.7	0.8–3.7	0.205
*RB1*	0.9	0.5–1.6	0.803	*RB1*	1.2	0.6–2.2	0.645
*BCL2*	1.5	0.8–2.5	0.287	*BCL2*	2.6	1.4–4.8	0.004
*ATM*	1.1	0.7–1.9	0.667	*ATM*	1.0	0.6–2.0	0.921
*TP53*	2.3	1.4–3.6	0.001	*TP53*	2.2	1.2–3.8	0.008
*CDKN2A*	2.6	1.5–4.7	0.001	*CDKN2A*	2.5	1.2–4.9	0.011
*MYC*	1.6	1.0–2.6	0.06	MYC	1.2	0.7–2.2	0.507

Tables show Cox’s proportional hazard model; HR = hazard ratio, CI = confidence interval *p* = *p*-value; statistically significant results are highlighted in gray.

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
