# Peer review of "Concurrent TP53 and CDKN2A Gene Aberrations in Newly Diagnosed Mantle Cell Lymphoma Correlate with Chemoresistance and Call for Innovative Upfront Therapy"

_cancers, 2020, doi:10.3390/cancers12082120_

Round 1
Reviewer 1 Report
The authors evaluated the prognostic impact of seven recurrent gene aberrations including ATM, CDKN2A, BCL2, MYC, RB1, CDK4 and TP53 in a cohort of 126 patients with MCL using FISH and NGS. They reported a novel molecular/cytogenetic index combining TP53 deletion/mutation and CDKN2A deletion that allow to identify high-risk MCL patients that will .
This is an interesting study, including a significant number of patients, with clinical potential to identify high-risk MCL patients that will be resistant to immunochemotherapy-based treatment. Several comments could be addressed to strengthen the manuscript.
Comments:
1/ The authors should include and discuss this recent study in their manuscript:
Streich L et al. Aggressive morphologic variants of mantle cell lymphoma characterized with high genomic instability showing frequent chromothripsis, CDKN2A/B loss and TP53 mutations: a multi-institutional study. Genes, Chromosomes and Cancer. 2020.
2/ It could be interesting to know if subclonal heterogeneity could be identified and the link with outcome in the patients.
3/ The authors should discuss how the risk stratification could provide potential targets for alternative therapies/targeted therapies.
Author Response
Thank you for an overall positive review and please find attached our responses to your valuable comments. We were, however, not able to re-submit the revised version of the manuscript in the on-line submission system. We have contacted the editorial office to fix this issue.
Best.
Pavel Klener, corresponding author

Reviewer 2 Report
Review of “Concurrent TP53 and CDKN2A Gene Aberrations in Newly Diagnosed Mantle Cell Lymphoma Correlate With Chemoresistance and Call for Innovative Upfront Therapy” by Diana Malarikova et al.
This is a solid analysis of the impact of genetic aberrations in 7 genes on the prognosis and response to chemotherapy of human patients with mantle cell lymphoma. In particular, the central analyses by FISH and ultra-deep next-generation sequencing of TP53 and CDKN2A has apparently been performed in a technically very solid and reliable way. All relevant results have been made publicly accessible, which is very positive. However, the presentation of Figures and Tables and the writing style still leaves room for improvement.
This reviewer suggests the following major and minor revisions which should help to improve the paper further.
Major points
- The authors frequently use “pts” as an abbreviation for “patients”. This reviewer suggests to use “patients” instead of “pts” throughout. “pts” is not much shorter anyway (by only 5 letters), is kind of disrespectful towards patients, and is less clear and more difficult to read.
- The authors refer to their cohort as “real-word cohort”, this should be omitted and replaced just by “cohort”, since also study cohorts are clearly part of the real world.
- 2.1. Patient characteristics: It is stated that FISH was performed between January 1, 2009 and June 30, 2018, and that the study was approved by University General Hospital Ethics Committee 48/18. Assuming that 48/18 is the number of the Ethics Committee Approval and 18 refers to the year 2018, there seems to be a temporal discrepancy. Presumably, the FISH analyses were performed in 2018, but the samples were collected 2009-2018. This should be clarified.
- The section 5 “Conclusions”, which is mandatory according to the Instructions to Authors, is completely missing from this manuscript.
Minor points
- The term “Legend:” can be omitted from the legends to Figures and Tables, it is obvious from the position directly underneath the labelling “Figure 1” etc. that it is the legend.
- Page 2, line 4: “reliable marker of true chemoresistance” should be replaced by “reliable marker of chemoresistance”
- Legend to Table 1: the abbreviation “dg.” Is not explained in the legend, unlike all other abbreviations.
- Figure 2: The quality of the graphics is rather poor, it is difficult to discern which legend refers to which curve. The graphics are better in Figure 4, use this format instead.
- Figure 3: The legend indicates that this figure shows the aberrations of 101 patients. However, this Figure has only 52 lines, and all of them have TP53 aberrations, so presumably the figure shows only the subset of patients who do have TP53 aberrations. Moreover, the legend indicates that “gray squares represent aberrated genes”, however, there seems to be simply every second line in gray, irrespective of any genetic aberrations. In addition, it is indicated that some patients have both a monoallelic deletion (“2”) and a biallelic deletion (“3”) of CDKN2A simultaneously, which is a contradiction. Please clarify.
- Table 2: The legend indicates that “Statistically significant results ae highlighted in gray”, however, there is nothing highlighted in gray. Moreover, the legend needs more detail, what is the statistical parameter and method shown in Table 2? (This reviewer assumes it is p-values of Chi-square tests, but this should be indicated in the legend).
- Page 7, last paragraph: The sentence “Because TP53 deletion and mutation significantly correlated one with another concerning survival parameters (EFS and OS), all analyses were implemented using TP53 aberration (i.e. TP53 mutation and/or deletion)” is unclear. This reviewer suggests to rephrase, e.g. “Because TP53 deletion and mutation significantly correlated with each other with respect to survival parameters (EFS and OS), we used TP53 aberration (defined as TP53 mutation and/or deletion) for all analyses”
- Page 8., line 2: The sentence “Likewise, a total number of FISH aberrations (including TP53 mutation) also negatively correlated with EFS and OS.” is unclear and should be rephrased. Either the total number of FISH aberrations…correlates with EFS and OS, or a high number of FISH aberrations…negatively correlates with EFS and OS.
- Figure 4 should be labelled as “Figure 4”, currently it is only labelled as “Legend:”. Moreover, Figure 4 has panels A, B, B and D, but no C. The second “B” should be replaced by “C”.
- Table 3 needs a legend, which should indicate what statistical parameters and methods are shown and what the highlights in gray mean. Moreover, the abbreviations “HR”, “CI” and “p” should be explained.
- The section “Supplementary Materials” should be completed in a meaningful way, currently it shows just the text of the manuscript template of Cancers.
- The Manuscript has both a section “Author Contributions” (which shows just the text of the manuscript template of Cancers) and “Authorship” (which seems to refer to the current manuscript). These two sections should be fused in a meaningful way and according to the Instructions to Authors of Cancers.
Author Response
Thank you for an overall positive review and please find attached our responses to your valuable comments. We believe that some of your minor points related to technical issues associated with file transfer (missing parts of figures or text). In addition, we were not able to re-submit the revised version of the manuscript in the on-line submission system. We have contacted the editorial office to fix this issue.
Best.
Pavel Klener, corresponding author
